# Calcium Nutrition in Fig Orchards Enhance Fruit Quality at Harvest and Storage

Jackson Mirellys Azevêdo Souza [1], Sarita Leonel [2,*], Magali Leonel [3], Emerson Loli Garcia [3], Luiza Rocha Ribeiro [2], Rafael Bibiano Ferreira [2], Rafaelly Calsavara Martins [2], Marcelo de Souza Silva [4], Laís Naiara Honorato Monteiro [2] and Anita Santos Duarte [2]

[1] Department of Agronomy, Federal University of Viçosa (UFV), Viçosa 36570-900, MG, Brazil
[2] Department of Horticulture, School of Agriculture, São Paulo State University (UNESP), Botucatu 18610-034, SP, Brazil
[3] Center for Tropical Root and Starches, São Paulo State University (UNESP), Botucatu 18610-034, SP, Brazil
[4] Faculty of Higher Education and Integral Training, FAEF, Garça 17400-000, SP, Brazil
* Correspondence: sarita.leonel@unesp.br; Tel.: +55-14-3880-7503

**Abstract:** Due to the high perishability of figs, calcium stands out as an important nutrient for orchard management. This study aims to study the pre-harvest applications of calcium chloride in fig orchards. The fig trees were sprayed with 0, 0.5, 1.0, 1.5 and 2.0% $CaCl_2$, and fruits were thereafter stored for 0, 7, 14 and 21 d. The variables analysed were the concentration of calcium in leaves and fruits, weight loss, fruit decay, pectinolytic enzyme activity, and physical and chemical attributes of the fruits; $CaCl_2$ sprays enhanced fruit $Ca^{2+}$ concentration in leaves (14.03%) and fruits (29.3%) and were effective in reducing polygalacturonase and pectin methylesterase activity, weight loss and fruit decay. Pre-treatment with 1% $CaCl_2$ provided fruits with larger diameters, greater firmness, and higher levels of total phenolic compounds in both fruit peel and pulp. The pre-harvest application at 1% $CaCl_2$ showed to be a promising technique in producing high quality fruits and extended storage by approximately 7 to 10 days. These findings may be useful in planning new cultural practices for fig orchards that produce high-quality fruit with desirable characteristics for growers and consumers.

**Keywords:** *Ficus carica* L.; fruit decay; pectin methylesterase; polygalacturonase; weight loss

## 1. Introduction

Historically, ancient cultures have exploited figs as a cultural food, and figs are fruits full of health benefits [1]. The Mediterranean basin has long been a site of fig production, especially Egypt, Turkey and Algeria. Brazil is the 11th largest producer in the world, accounting for 19.6 thousand metric tonnes in 2020 [2]. The cultivar 'Roxo de Valinhos' is the most used by Brazilian producers; the fruits are extremely perishable, though. This perishability is due to the fig being a climacteric fruit with rapid physiological breakdown and thin skin that makes it susceptible to pathogen attack [3].

Pre-harvest spraying with calcium in fig orchards is a promising cultural practice. Calcium-pectin cross-links promote the stabilisation of cell wall structures and limit their degradation by the action of enzymes [4], such as pectin methylesterase (PME) and poly-galacturonase (PG). These pectin-degrading enzymes are the most implicated in fruit-tissue softening, wherefore the exogenous application of calcium helps stabilise the plant cell wall by maintaining tissue firmness and reducing weight loss in fruits [5].

Furthermore, some studies have shown that calcium applications have reduced the incidence of microorganisms in various fruits [3,6,7]; others have reported the involvement of calcium in phenolic compound synthesis and antioxidant enzyme activity [8–10]. The concentration of phenolic compounds is important, as consumer trends drive demand for functional foods. Thus, the likely increase in these substances makes fig a great diet for

humans, since phenolic compounds can help preventing and/or treating several diseases when vegetables and fruits are consumed [11].

However, calcium is poorly phloem-mobile; therefore, the accumulation of calcium in fruits depends on the water that is pulled up through the xylem by the force of transpiration, but the fruits have low rates of transpiration when compared to leaves [12]. This results in low calcium accumulation in the fruits, making it necessary to adopt techniques that will meet the need for this mineral, especially in trees with vigorous vegetative growth, such as fig trees.

Alternatively, calcium can be supplied to the figs by immersing the fruits in $CaCl_2$ solution up to 4% after harvest. Moreover, considering that figs are highly perishable fruits, alternative methods to immersion, such as pre-harvest spraying, avoid handling and possible damage to fruits. There are no reports of pre-harvest spraying of calcium on figs, but studies have shown promising results in other fruits, such as blueberries [10], guavas [13], blackberries [14] and pears [4]. In Californian-style black and green olives, the effect of calcium strategies by immersion in the fruit is well known [15]. However, soaking olives in calcium may provide some negative bitterness attributes provided by this mineral [16]. These reports [15,16] indicated that $CaCl_2$ is a firming agent, which is often added to the package table olive top brine to improve olive firmness due to calcium ions that help maintain structural firmness and cell wall stability and the cell turgor of fruits that form cross-links between pectin molecules. This strengthens the plant cells and prevents this breakdown.

Importantly high $Ca^{2+}$ concentrations in fruits result in cellular toxicity and abnormalities in plant development. Large amounts of calcium can move from vacuoles and intercellular spaces to the cell cytoplasm, triggering greater activity of the ascorbate-glutathione cycle and consequently promoting greater synthesis of ascorbic acid. This process is a reflection of the plant's protection system against oxidative stress. Ascorbic acid acts as an antioxidant in the plant, regulating the amounts of reactive oxygen species (ROS) that contribute to injuries resulting from stress, causing cell damage, which are visible on leaves in the form of lesions, especially when excess calcium is applied [12].

The effects of pre-harvest applications of calcium chloride on fruit quality still have not been well investigated. There is a shortage of information regarding the source, the concentration, the number, and the period of calcium application onto the trees to get an effective response to fruit quality [8,9]. Therefore, studies that come to enable the adoption of safety concentrations by fig growers seeking the production of higher quality fruit. Consequently, this study aimed to find an appropriate calcium concentration for fig based on fruit quality and storage.

## 2. Materials and Methods

### 2.1. Characteristics of the Experimental Area

The experimental field was installed at the farm for the School of Agriculture, São Paulo State University (UNESP), located in the city of São Manuel, São Paulo (22°44′28″ S, 48°34′37″ W at an altitude of 740 m). Based on the Köppen classification, the climate in this area is of the Cwa type (temperate hot—mesothermal). Rainfall is concentrated from November to April (summer season), and the average value of precipitation is 1376.70 mm. The temperature of January, the hottest month, is 22 °C. The soil is classified as a sandy-textured Latossolo Vermelho distroférrico according to the Brazilian system of soil classification [17], that is, a dystrophic Typic Hapludox [18].

The fig orchard was established in June 2013 by planting the fig tree (*Ficus carica* L. cv Roxo de Valinhos) at 3 m spacing between rows and 2 m between trees. The correction of the soil and fertilisation of the entire orchard were based on previous soil analyses and crop recommendations. One hundred and twenty trees were conducted in a rain-fed system, receiving all the standard cultural practices recommended for the crop. Twenty guard trees were used to preserve some treatments from others.

### 2.2. Pre-Harvest Applications of Calcium

Calcium chloride (CaCl$_2$) with 27% of Ca$^{2+}$ was used as a source of calcium in this study. CaCl$_2$ was applied with a backpack sprayer at 0; 0.5; 1.0; 1.5 and 2.0%. The product was diluted with water and adhesive spreader (2 mL L$^{-1}$—Assist$^®$, São Paulo, Brazil). The first CaCl$_2$ application was performed as soon as fruit emission began, that is, October 2019. Then, 10 applications were made at 15 d intervals during the 2019 and 2020 harvest seasons. Calcium chloride was applied when the fruit setting began and finished in a ripening stage less unripe, with firmer fruits [19]. Five hundred millilitres of solution per tree were used per application at ambient temperature. The sprayings were directed towards the entire aerial part of the trees (i.e., leaves and mainly fruits).

### 2.3. Harvest and Storage of Figs

The fruits were harvested when at least 50% of their surfaces had a red-purple colour. After harvesting, they were cleaned in running water and placed on benches for natural drying. Subsequently, the fruits were placed in polystyrene trays covered with polyvinyl chloride (PVC) film and were therefore placed in a cold chamber (5 ± 1 °C; 90–95% R.H.) for 0, 7, 14 and 21 storage days.

### 2.4. Calcium Concentration in Leaves and Fruits

The levels of calcium in the leaves and fruits were measured soon after harvest. Samples were washed with deionised water to avoid contamination and were dried in an oven under forced-air ventilation at 60 °C until the weight remained constant, then ground in a Willey mill. Calcium concentration was determined according to Malavolta et al. [20]. After sample digestion, the calcium had the absorbance measured in an atomic absorption spectrophotometer at 422.2 nm, expressed in g kg$^{-1}$.

### 2.5. Weight Loss and Fruits Rot Decay

At fruit packing time, trays were randomly selected and identified for the evaluation of weight loss and fruit rot decay (conidia of *Aspergillus* fungi). For weight loss, the same fruits were used for the initial weight and the final weight. The trays were weighed on an analytical balance during storage (0, 7, 14 and 21 days) and the difference between the initial and final weight of each interval was calculated. The following equation was used: Weight loss (%) = [(WI − WF/WI)] × 100, where WI is the initial weight (g) and WF is the weight on the evaluation day (g).

The percentage of fruit rot decay was assessed by means of the presence or absence of pathogenic rot decays or peel spots (lesion diameters greater than 3 mm). Results were expressed as the percentage of decayed fig fruits. Evaluations were performed at 2-day intervals from day 0 (harvest) to the 22nd day of storage. The number of fruits per replicate was therefore taken to calculate the percentage of decayed fruits by using the following equation: % decayed fruits = [(number of decayed fruits/total number of fruits)] × 100, according to the methodology described by Shiri et al. [5].

### 2.6. Degrading Enzyme Activity

Samples from each treatment corresponding to the CaCl$_2$ concentrations (plot) at each storage time (subplot) were sliced and ground in a pestle and mortar with liquid nitrogen. The action of the enzyme's polygalacturonase (PG) and pectin methylesterase (PME) was measured using the protocol introduced by D'Innocenzo and Lajolo [21]. The extraction was performed by diluting 5 g of the sample in 5 mL of buffer with pH 5.0 (12% polyethylene glycol + 0.2% sodium bisulfide). Then, the extract was homogenised in turrax (IKA T10—90 s, Staufen, Baden-Würetmberg, Germany) and centrifuged (Helttich Mikro 220 R—24,510 g, 30 min, 4 °C, St. Louis, MO, USA). The recovered pellet was homogenised with sodium acetate (0.1 M) buffer (pH 4.5) with NaCl (0.1 M) in an ice bath under constant agitation. The extract was again centrifuged (12,255 g, 20 min, 4 °C) and the supernatant

was dialysed in 0.1 M sodium acetate buffer (pH 4.5) with 1.0 M NaCl. This dialysate was used as a source of enzymes.

### 2.7. Fruit Firmness, Size and Colour

The firmness, size and colour analyses of the figs were performed at 0, 7, 14 and 21 d of storage. For storage time evaluation, 32 figs (4 trays of 8 fruits) were evaluated for each $CaCl_2$ concentration. For firmness, the penetration resistance evaluated the texture of fig fruit, and the measurements were performed using a TA.XT Plus Texture Analyzer equipped with a cylindrical probe (SMS P/2N). Tests were performed in two equatorial regions of the fruits. The penetration depth was set at 5 mm (50 mm s$^{-1}$), results were expressed in Newton (N). The size of the fruits was measured by their diameter and length, measured with a digital vernier calliper (Starrett 799A-6/150, Athol, MA, USA) and expressed in millimetres (mm). For colour parameters, fruit colour was evaluated with a *CR-400* Chroma Meter (Konica Minolta, Osaka, Japan). The colour changes were evaluated using the CIELAB colour space. The peel colour was measured on opposite sides of the mid-equatorial region of each fruit surface. Chromaticity (C) and hue angle (°h) were evaluated.

### 2.8. Chemical Evaluation of Fruits

The same fruits used for physical properties were used for chemical analyses, also performed on the day after harvest, 0 d and at 7, 14 and 21 d. The samples were crushed with the aid of mixed fruit (Philips Walita—RI1364, Varginha, Minas Gerais, Brazil) to form a homogeneous extract of the figs. However, for the analysis of total phenolic compounds and antioxidant activity, the preparation of the samples differed. The figs were sliced and frozen in liquid nitrogen and then ground with the aid of a mortar and pestle.

Titratable acidity (TA) was achieved by titrating with sodium hydroxide (NaOH— 0.1 N) in a solution of 1 g of the homogenised fig extract, 50 mL of distilled water and 0.3 mL of phenolphthalein [22], expressed as % of citric acid. Soluble solid content (SSC) was measured with a digital refractometer (Atago 3405 PR-32a, Minato-ku, Tokyo, Japan) using 1 g of the homogenised fig extract and expressed as %. The SS/TA ratio was also measured, which indicates the fruit ripeness index (expression of sweet taste).

Reducing, non-reducing and total sugars were evaluated using the phenol-sulphuric method [18] and the DNS method (3,5-dinitrosalicylic acid) [23,24]. One gram of the homogenised fruit extract diluted in 100 mL of distilled water was used. The evaluations were performed using a UV-visible spectrophotometer (BEL photonics®, Piracicaba, São Paulo, Brazil), with readings at a wavelength of 490 and 540 nm and results expressed as percentage (%).

The same extract was used for total phenolic compounds and antioxidant activity. Samples (100 mg) were diluted in 3 mL of pure methanol and kept in an ultrasonic bath for 30 min. Then, the samples were centrifuged (Helttich Mikro 220 R—24,510 g, 15 min, 4 °C) and the supernatant was collected for analysis.

The total phenolic concentration in the peel and pulp of the fruits was determined using the Folin–Ciocalteau colorimetric method [25]. The analyses were performed on a UV-visible spectrophotometer (BEL photonics®, Brazil) at 765 nm and the absorbance was compared with a calibration curve for gallic acid, expressed as mg 100$^{-1}$ gallic acid equivalents (GAE).

Antioxidant activity was evaluated in the peel and pulp of the fruits by a method based on the use of the DPPH reagent (2,2-Diphenyl-1-picrylhydrazyl) [26]. Readings were taken at 517 nm on a UV-visible spectrophotometer (BEL photonics®, Brazil). From the absorbance, the scavenging activity was calculated and hence they were compared on a calibration curve obtained for Trolox. The results were expressed in mg 100 g$^{-1}$ in Trolox equivalent antioxidant capacity (TEAC).

*2.9. Experimental Design and Statistical Analysis*

Randomised complete block design was adopted with subdivided plots, that is, plots corresponding to the $CaCl_2$ concentrations (0, 0.5, 1.0, 1.5 and 2.0%) and the subplots to the storage time (0, 7, 14 and 21 d). In the field, each plot was composed of four replicates, with five trees each, totalling 100 useful trees. In storage, each subplot was composed of four replicates of 10 fruits each, totalling 40 fruits per storage time, 160 by $CaCl_2$ concentration and 800 total fruits.

The analysis was based on average data from the 2019 and 2020 seasons. The normality hypothesis was tested by the Shapiro–Wilk test, and the F test (1 and 5% levels of significance) was used in the analysis of variance to detect the difference between the factors. When a significant difference was found for the sources of variations, regression analysis (linear and quadratic) was used to compare the treatment means for the data of the plots ($CaCl_2$ concentration) and subplots (storage time). To ensure the homogeneity of variances in the regression analysis, Grubb's test was applied to verify the presence of outliers for each concentration level.

## 3. Results

### 3.1. Calcium Concentration in Leaves and Fruits

Pre-harvest $CaCl_2$ spray significantly increased in calcium concentration in treated leaves ($p < 0.01$) and fruits ($p < 0.01$). The calcium concentration in the leaves of Roxo de Valinhos figs increased linearly up to the concentration of 2.0% $CaCl_2$, when they reached 18.3 g kg$^{-1}$ of calcium, corresponding to an increase of 14.03% in relation to the control (Figure 1a). Similarly, the Ca concentration in fruits increased linearly with the application of 2.0% $CaCl_2$, reaching a value of 7.15 g kg$^{-1}$, that is, and an increase of 29.3% in relation to untreated trees (Figure 1b). However, the leaves presented toxicity symptoms at 1.5 and 2.0% $CaCl_2$ (Figure 2).

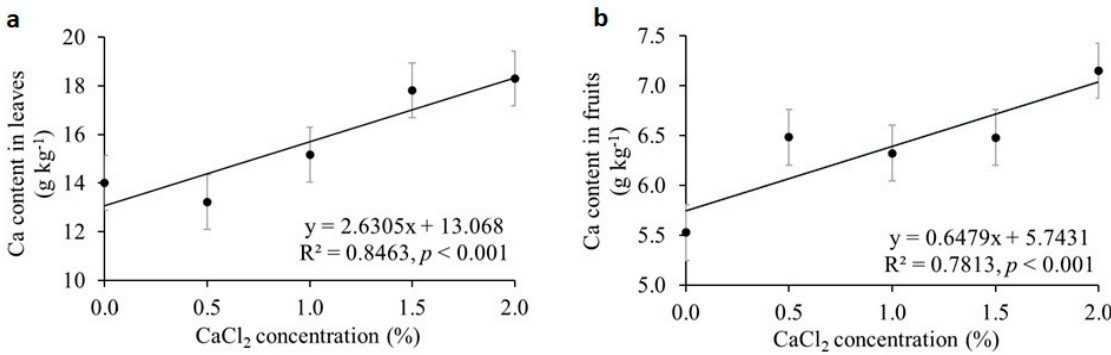

**Figure 1.** Calcium concentration in leaves (**a**) and fruits (**b**) of fig tree cv. 'Roxo de Valinhos' as a function of different concentrations of pre-harvest $CaCl_2$ spray. Bars in each column symbolise standard errors (n = 4).

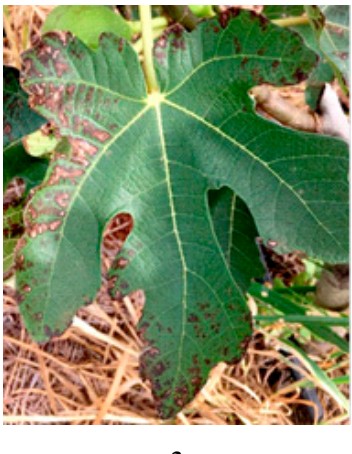

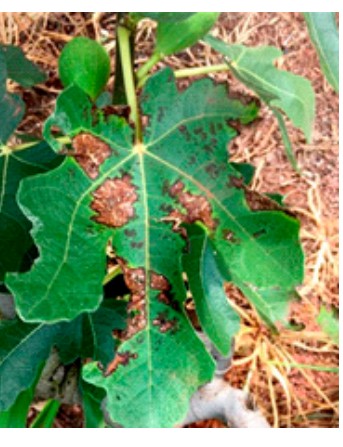

**a**　　　　　　　　　　　　　　　　　　**b**

**Figure 2.** Symptoms of $Ca^{2+}$ toxicity in fig leaves cv. 'Roxo de Valinhos' as a function of pre-harvest $CaCl_2$ spray at 1.5% (**a**) and 2.0% (**b**).

### 3.2. Weight Loss and Fruit Decay

Significant differences were found for the single effect of $CaCl_2$ concentrations ($p < 0.01$) and storage ($p < 0.01$) on the figs weight loss. When weight loss was evaluated upon $CaCl_2$ concentrations, there was a quadratic decrease of up to 0.9% $CaCl_2$ (value obtained by the fit curve), reaching an average of 1.8%, equivalent to a reduction of 20.7% (Figure 3a). Regarding the storage days, weight loss evolved linearly, reaching the highest value at 21 d (Figure 3b).

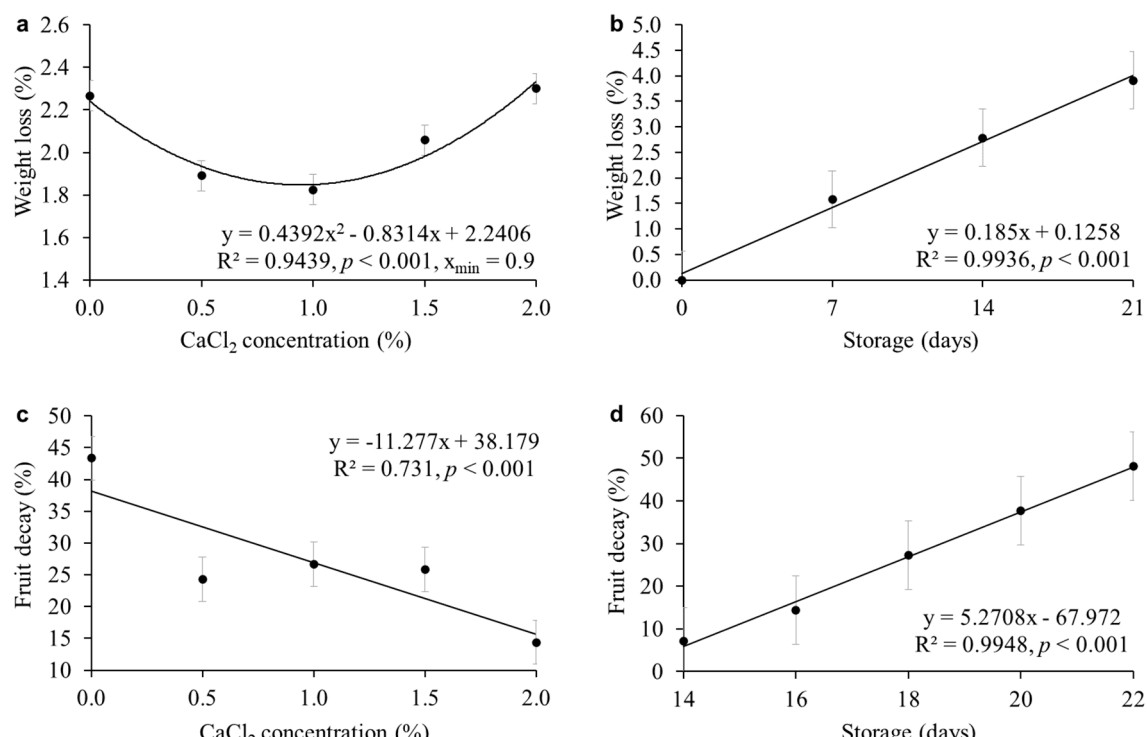

**Figure 3.** Weight loss (**a**,**b**) and fruit decay (**c**,**d**) in 'Roxo de Valinhos' fig fruits as a function of $CaCl_2$ concentrations and storage days at $5 \pm 1$ °C and 90–95% R.H. Bars symbolise standard errors (n = 4).

Moreover, a significant difference was also found for the effect of the concentrations ($p < 0.05$) on the decay percentage of fig fruits, whose averages decreased linearly with an increase in $CaCl_2$ percentage, resulting in a 66.8% reduction in the decay percentage of fig

fruits (Figure 3c). However, fruit decay increased linearly with an increase in storage days ($p < 0.01$). The first decay symptoms appeared only after 14 d of storage, regardless of the CaCl$_2$ concentration (Figure 3d).

### 3.3. Degrading Enzyme Activity

There was a significant effect of the interaction between CaCl$_2$ concentrations and storage for the PME ($p < 0.01$) and PG ($p < 0.01$) activities. Soon after harvest (day 0), high PME activity was observed at 2.0% CaCl$_2$. However, there was a quadratic increase in the averages of untreated and treated trees with 0.5% CaCl$_2$ throughout storage time. In general, PME activity remained low at 1.0 and 1.5% CaCl$_2$ (Figure 4a). Similarly, high PG activity was obtained in the untreated and treated trees with 2.0% CaCl$_2$. Therefore, PG activity was fit to a quadratic regression model throughout storage, and the lowest averages were mainly obtained at 1.5% CaCl$_2$ (Figure 4b).

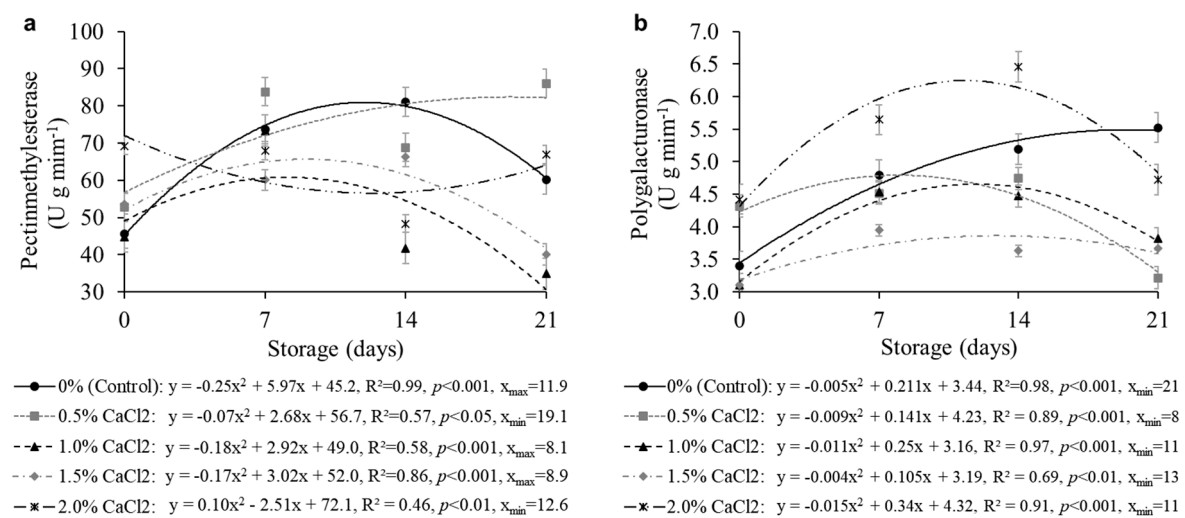

**Figure 4.** Pectin methylesterase (**a**) and polygalacturonase (**b**) activities of 'Roxo de Valinhos' fig fruits as a function of CaCl$_2$ concentrations and storage days at $5 \pm 1$ °C and 90–95% R.H. Bars symbolise standard errors (n = 4).

### 3.4. Fruit Firmness, Size and Colour

Significant differences were also found for the single effects of CaCl$_2$ concentrations ($p < 0.01$) and storage ($p < 0.01$) on fruit firmness. The applications promoted a quadratic increase in fruit firmness up to 1.0% CaCl$_2$ (Figure 5a). The firmness of the untreated fruits was 0.28 N, while those that received about 1.0% CaCl$_2$ was 0.45 N, that is, an increase of 59.8%.

However, there was a linear reduction in fruit firmness from 0.48 N to 0.28 N throughout storage, that is, a decrease of 42.4% (Figure 5b). Fruit length was not affected by CaCl$_2$ concentrations and storage, or even by the interaction of these factors, so the average fruit length was 61.9 mm. On the contrary, the fruit diameter was significantly affected by CaCl$_2$ concentrations ($p < 0.05$) and storage ($p < 0.01$). There was a quadratic increase in fruit diameter up to 1.0% CaCl$_2$ (0.8% by regression test) (Figure 5c). Regarding storage days, there was a linear reduction in the diameters of the fruits (Figure 5d).

For chromaticity ($p < 0.01$) and hue angle ($p < 0.01$), there was a significant effect of the interaction between CaCl$_2$ concentrations and storage. During storage, the only increase in the chromaticity of the fruit surface was observed at 1.0% CaCl$_2$ concentration, while the other concentrations (0.5, 1.5 and 2.0% CaCl$_2$) presented a linear reduction. There was a small increase in chromaticity up to 6 days of storage in the untreated trees, but also a reduction after this period (Figure 5e).

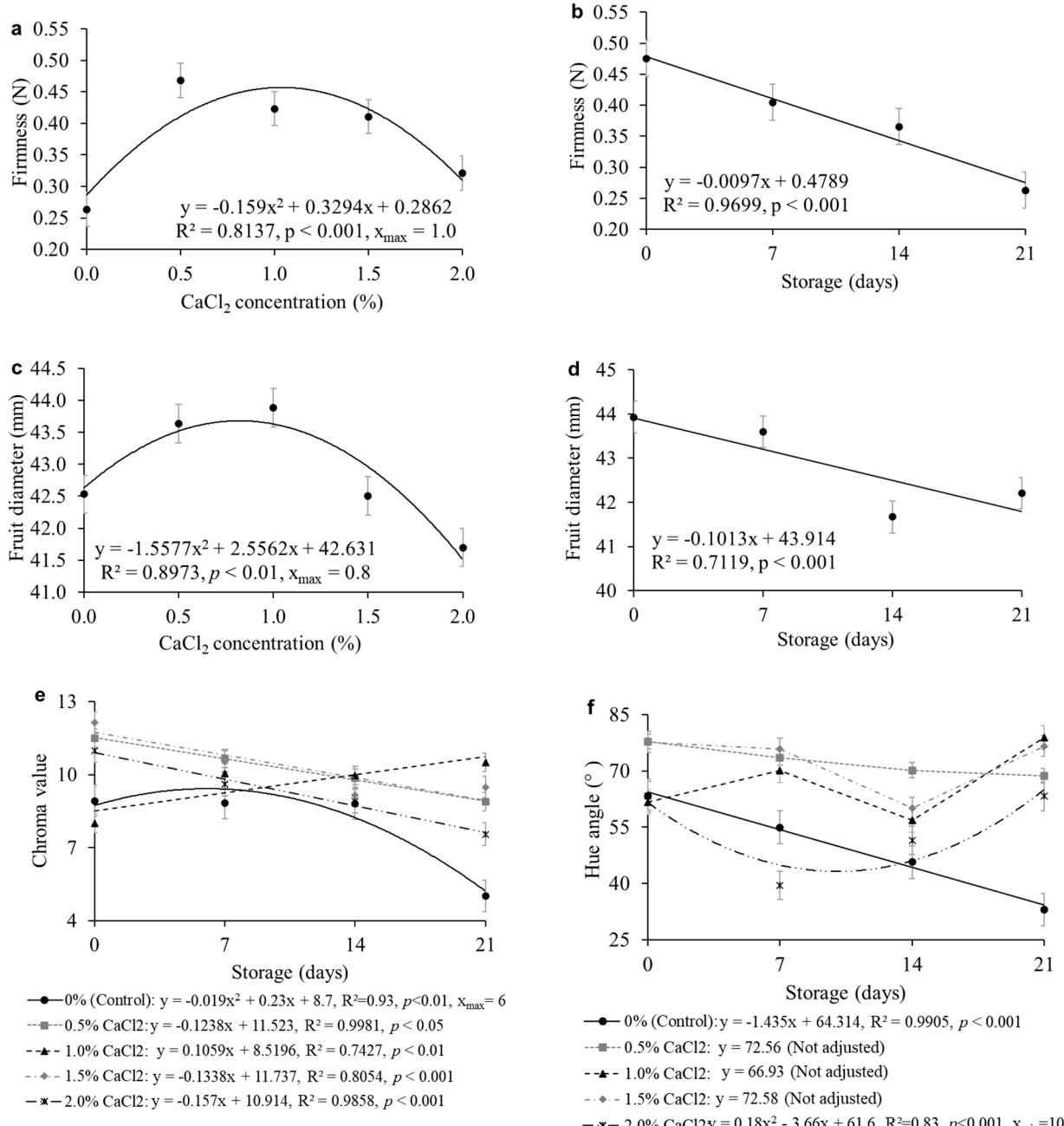

**Figure 5.** Firmness (**a**,**b**), diameter (**c**,**d**), chroma value (**e**) and hue angle (**f**) of 'Roxo de Valinhos' fig fruits as a function of $CaCl_2$ concentrations and storage days at $5 \pm 1$ °C and 90–95% R.H. Bars symbolise standard errors (n = 4).

The untreated trees had a linear decrease in the hue angle values, while there was a reduction until 10 days of storage, with a subsequent increase until 21 days in the trees treated with 2.0% $CaCl_2$. Regarding the other concentrations, there was no adjustment of the evaluated equations (linear and quadratic). This means that for this variable, there was no regular pattern of response. However, it appears that the hue angle obtained with the intermediate concentrations (0.5, 1.0 and 1.5% $CaCl_2$) was higher than the others (Figure 5f).

*3.5. Chemical Quality*

The interaction between $CaCl_2$ concentrations and storage time was significant for soluble solids content ($p < 0.05$). Fruits treated with 1% $CaCl_2$ showed a higher content of soluble solids on the day of harvest (day 0). However, over storage time, there was a

quadratic reduction of soluble solids for concentrations of 0 to 1.5% of CaCl$_2$ and linear for the concentration of 2.0% (Figure 6a). The highest means of soluble solids at 21 days were found in the untreated fruits.

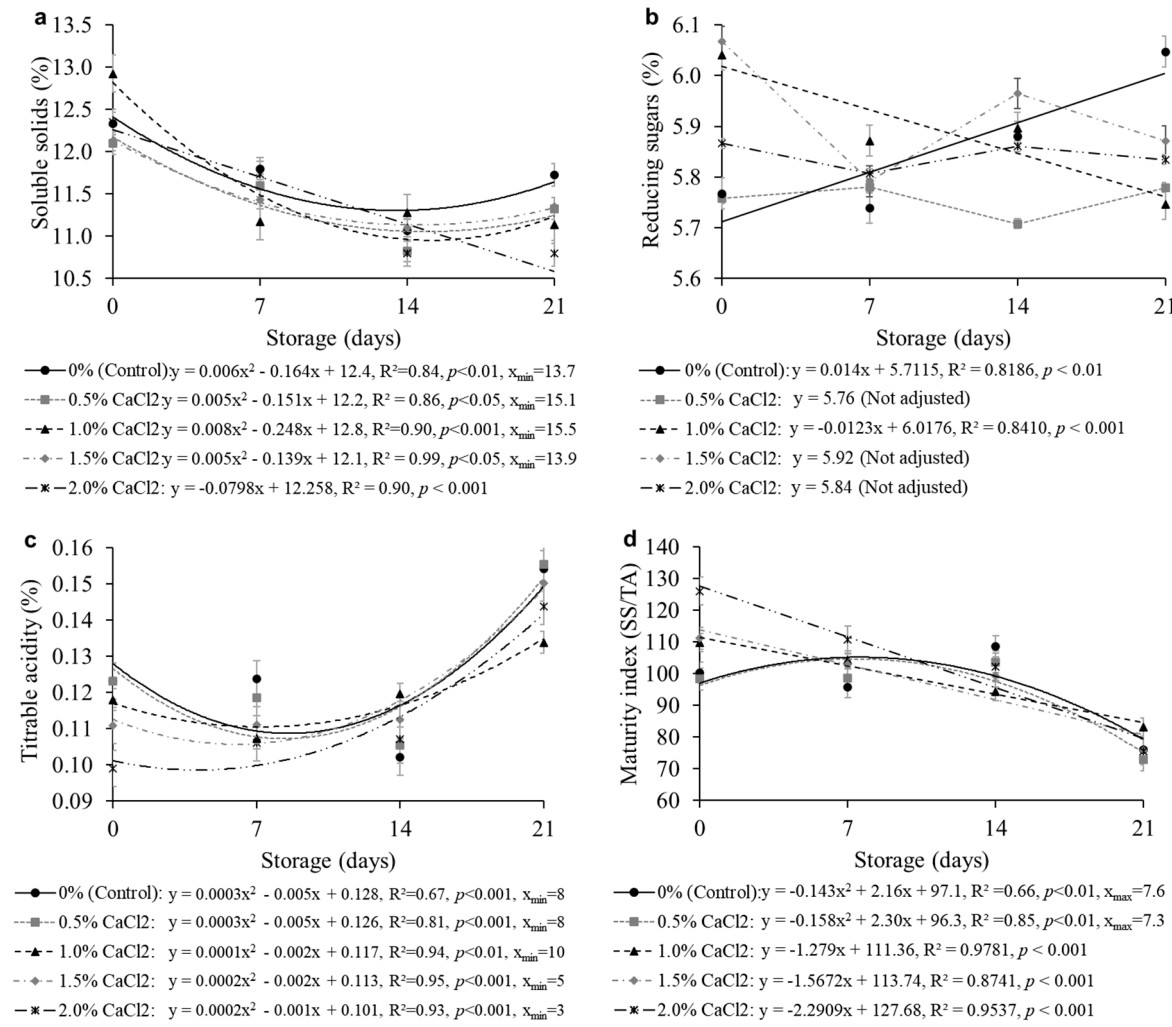

**Figure 6.** Soluble solids (**a**), reducing sugars (**b**), titratable acidity (**c**) and maturation index (**d**) of 'Roxo de Valinhos' fig fruits as a function of CaCl$_2$ concentrations and storage days at 5 ± 1 °C and 90–95% R.H. Bars symbolise standard errors (n = 4).

The contents of reducing sugars were significantly affected by the interaction between concentrations and storage ($p < 0.01$). While there was a linear increase in reducing sugars during storage in the untreated trees, a linear reduction was observed in the trees treated with 1.0% CaCl$_2$. Furthermore, the means of reducing sugars of the other concentrations did not adjust to the applied equations (Figure 6b). The contents of non-reducing and total sugars were not affected by CaCl$_2$ applications or even by storage, with averages of 0.8 and 6.7%, respectively.

There was a significant effect of the interaction between CaCl$_2$ concentrations and storage ($p < 0.01$) for titratable acidity, since there was an initial reduction in the averages with a subsequent sharp increase in all treatments, reaching the highest values at the end of the 21d storage period. Moreover, untreated and treated trees with 0.5% CaCl$_2$ showed the highest titratable acidity throughout storage (Figure 6c).

A high maturation index was found in fruit treated with 2.0% CaCl$_2$ at the beginning of the storage period due to the reduced titratable acidity of these fruits. The maturation index decreased linearly throughout storage when concentrations were applied between 1.0 and 2.0%. Similarly, data was fit in a quadratic regression model in untreated and treated

trees with 1.0% $CaCl_2$, as the maturation index increases followed by a reduction until 21 d (Figure 6d).

There was a significant effect of the interaction between $CaCl_2$ concentrations and storage for the total phenolics concentration of the peel ($p < 0.05$) and pulp ($p < 0.01$). In the peel, the highest concentration of total phenols was obtained, with 1.0 and 2.0% $CaCl_2$ at harvest (day 0) (Figure 7).

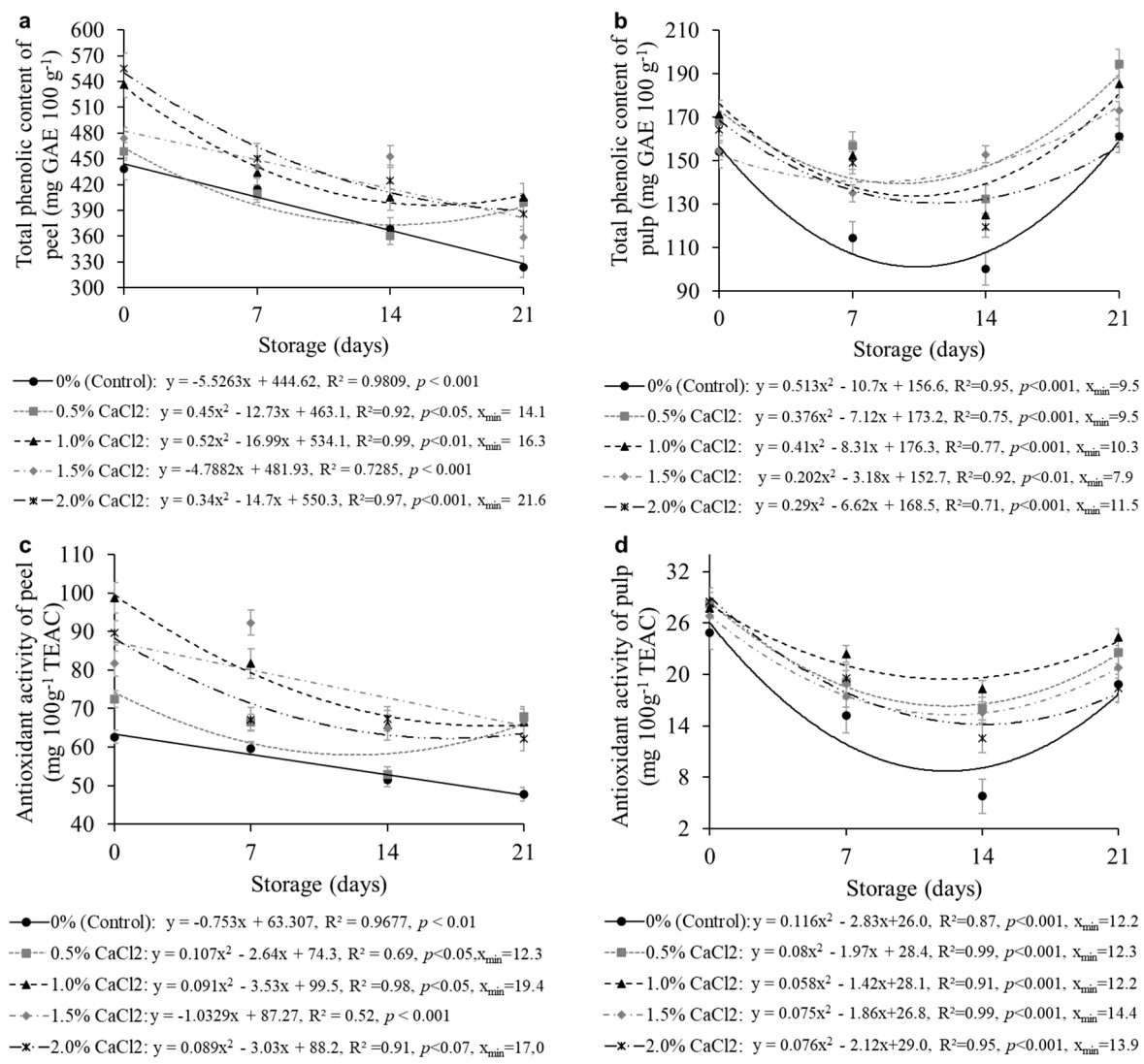

**Figure 7.** Total phenolic content of peel (**a**) and pulp (**b**) and antioxidant activity of peel (**c**) and pulp (**d**) of 'Roxo de Valinhos' fig fruits as a function of $CaCl_2$ concentrations and storage days at $5 \pm 1$ °C and 90–95% R.H. Bars symbolise standard errors (n = 4).

Throughout storage, there was a reduction in these values, with the lowest in the untreated fruits at the end of the 21d storage period (Figure 7a). Similarly, the pulp had a lower concentration of total phenols in the untreated fruits throughout storage (Figure 7b). For antioxidant activity, there was a significant effect of the interaction between $CaCl_2$ concentrations and storage ($p < 0.05$ for peel and pulp). In addition, there was a certain relationship between phenol concentration and antioxidant activity of the peel and pulp of the figs since the means during storage were similar between them. There was less antioxidant activity in both the pulp and peel of the untreated fruits (Figure 7c,d). Throughout the whole storage time, there was great antioxidant activity in the pulp of fruits treated with 1.0% $CaCl_2$.

## 4. Discussion

Pre-harvest $CaCl_2$ sprays increased the calcium concentration in fig fruits; thus, a promising and strategic method that improves human nutrition through biofortification of figs, as it can supply dietary calcium deficiency. Previous studies have shown an increase in $Ca^{2+}$ in fruits with the exogenous application of calcium in grapes [27], papayas [28] and kiwifruits [5]. Calcium-fortified fruits have been important not only from a nutritional point of view but also because they involve several physiological processes. However, an excessive or deficient amount of calcium can lead to major disturbances in plants [29].

Calcium deficiencies can lead to membrane breakdown and/or cell wall failure in fruits [30], reinforcing the importance of an adequate supply for trees. On the contrary, if tissue calcium concentration is high, this can result in cellular toxicity and developmental abnormalities [12], explaining the necrosis observed in fig leaves treated with the highest concentrations of $CaCl_2$. Calcium is moved in the plant metabolism by the transpiration flow and part of the calcium reallocated in the fruits ends up being translocated to meristematic regions where there are higher transpiration rates, such as growing leaves. This fact highlights the need for a constant supply of this nutrient during fruit growth [13]. This report reinforced the need for the ten $CaCl_2$ sprays needed in this study to improve final fig fruit quality.

Based on the results obtained, it can be suggested that leaf calcium concentrations above 16 g kg$^{-1}$ were toxic for fig tree cv. Roxo de Valinhos due to symptoms of leaf necrosis. This record must be considered since excessive necrosis leads to a decrease in photosynthesis areas and negatively affects the development and production of trees.

Calcium plays an important role in changes in cell turgor pressure and cell elongation in fruits, in addition to being involved in auxin signalling and acidification of the cell wall, allowing its expansion, which can lead to increased fruit weight [13]. However, the reduction in averages during storage, which was observed in all treatments, is related to the increase in weight loss in this period.

The increase in the calcium concentration in the figs was crucial to reduce weight loss, especially at the estimated concentration of 1.0% $CaCl_2$ (0.9% by fit curve–regression test). This reduction is because $Ca^{2+}$ provides structural integrity to cellular membranes. Previous research reported this efficient technique in reducing weight loss in red guavas [13], blackberries [14], blueberries [31], apricots [32], apples [33], peaches [34] and kiwifruits [5]. Besides maintaining the cell wall structure, calcium reduced the hydraulic conductivity of the fruits, which caused the lowest weight loss and is mainly due to water loss throughout storage [29]. However, the same authors reported that when excess in the cytoplasm, $Ca^{2+}$ causes damage to the membrane and increases respiratory rates. This explains the greater weight loss of figs when concentrations greater than 1.0% $CaCl_2$ are used.

Furthermore, the reduction in the percentage of decayed fig fruits can also be attributed to the maintenance of the cell wall structure promoted by $CaCl_2$ applications. In response to pathogen attack, there is a cytosolic influx of $Ca^{2+}$ followed by an apoplastic explosion of reactive oxygen species (ROS), promoting the activation of the antioxidant defence complex [29]. Irfan et al. [3] reported a lower incidence of aerobic bacteria, yeasts and fungi in figs cv. Poona treated with 4% $CaCl_2$ during post-harvest and, consequently, less fruit deterioration.

The efficiency of calcium in reducing polygalacturonase and pectin methylesterase activity has already been reported by Liu et al. [32] in apricot and by Ortiz, Graell and Lara [33] in apple cv. Golden Reinders. Pectin methylesterase promotes the de-esterification of pectin structure elements, forming a pectin with a lower degree of methylation; therefore, it is more susceptible to the adverse effects of polygalacturonase, which promotes the hydrolysis of (1–4) β-bonds between galacturonic acid residues present within the pectin chains [12]. However, the interaction between $Ca^{2+}$ and pectin forms a calcium-pectin gel called calcium pectate, which results in greater resistance and less degradation of cell wall components [4,29]. It is noteworthy that when in excess, the apoplastic $Ca^{2+}$ leakage causes damage to the cells, which explains the greater action of enzymes in fruits treated with 2.0%

of $CaCl_2$. Thus, it can be said that pre-harvest applications of $CaCl_2$ in safe concentrations promote lower pectin methylesterase and polygalacturonase activity, as well as greater fig firmness.

The great firmness of the fruits treated with 1.0% $CaCl_2$ occurred due to the calcium-pectin gel. $Ca^{2+}$ ions induce a chain-chain association, thus forming junction zones responsible for the formation of the gel. These zones are popularly known as the "egg box model." In this mechanism, initially two antiparallel polyuronate chains form the egg-box with $Ca^{2+}$ and further aggregate to form multimers [24]. Similarly, Lobos et al. (2020) [10] reported that early pre-harvest spraying of $CaCl_2$ increased blueberry firmness compared to untreated fruits.

High cytoplasmic concentrations of calcium cause damage to the membrane and increase the respiratory rate [29,33]; consequently, firmness decreases when concentrations are above 1.0% $CaCl_2$. On the other hand, the reduction of firmness is the result of cell wall hydrolases throughout storage, such as the enzymes pectin methylesterase and polygalacturonase, which alter the cell wall compositions [12].

Extracellular auxin promotes $Ca^{2+}$ transport in the plasma membrane, causing acidification of the cell wall solution that induces cell wall expansion. On the other hand, when in excess, the $Ca^{2+}$ ions compete with H+ for the same proton-binding sites, inhibiting cell wall acidification and growth [12]. This explains the larger diameter of fruit treated with an intermediate concentration of $CaCl_2$, as well as the smaller diameter of fruit treated with a high concentration (2%).

The maintenance of higher chroma and hue angle values obtained in fruits from trees treated with intermediate concentrations (i.e., 1.0% $CaCl_2$) may be due to chlorophyll retention, as calcium application reduces the respiratory rate and ethylene production in the fruits, with a consequent slowdown in the ripening phase [7,29]. The retention of green pigments is related to the accumulation of calcium in the cell wall and middle lamella, which promotes thickening of the fruit's cell wall due to the binding of $Ca^{2+}$ with protein molecules [3].

The large decrease in the concentrations of soluble solids and reducing sugar contents in fig fruits treated with $CaCl_2$ can be attributed to the major stabilisation of pectin bonds promoted by calcium [5]. The same authors reported that the soluble solid content in papaya decreased with increasing $CaCl_2$ concentrations. The effect of $CaCl_2$ treatments on titratable acidity presented variations between the evaluated concentrations and storage times. This variable also presented different responses in other fruit trees. Ribeiro et al. [13] reported that the pre-harvest application of $CaCl_2$ (0.4%) decreased the titratable acidity concentration in 'Paluma' red guava and Ali et al. [29] obtained results of high titratable acidity in peaches treated with 1.0% $CaCl_2$.

The high concentration of total phenols and antioxidant activity observed in figs treated with $CaCl_2$ are probably related to the action of $Ca^{2+}$. Calcium stimulates the activation of the NADPH oxidase to produce reactive oxygen species ($O_2$ and $H_2O_2$); thus, as a defence measure, there is a synthesis of phenolic compounds and great antioxidant enzyme activities formed by superoxide dismutase, peroxidase and catalase [8,29,34]. These findings were corroborated by Xu et al. [9], who reported that calcium treatment in strawberries increased anthocyanin accumulation and total phenolic concentrations. Similarly, Lobos et al. [10] reported the positive effects of $CaCl_2$ sprays on antioxidant activity and total phenol concentration in blueberries compared to untreated fruits.

There was a higher concentration of total phenols and, consequently, greater antioxidant activity of the peel in relation to the fruit pulp, regardless of $CaCl_2$ concentration or storage time. Similarly, Palmeira et al. [1] reported a higher concentration of phenolic compounds in the peel (152.0 mg 100 $g^{-1}$) compared to the pulp (54.2 mg 100 $g^{-1}$) in a study with Pingo de Mel figs. Therefore, consuming the peel with the pulp can boost the total intake of these nutrients, since some consumers only end up eating the pulp, especially in figs. Human diets composed of fruits and vegetables with great antioxidant activity act as barriers against cancer [11]. Therefore, the pre-harvest $CaCl_2$ application can be used

as an important technique for producing high-quality figs, since provided calcium-rich fig fruits, reduced weight loss and fruit rot decay, thereby enhancing the total phenolic concentrations and antioxidant activity.

In summary, the concentrations of $Ca^{2+}$ in the leaves and fruits of the fig tree increased linearly as a function of $CaCl_2$ applications. There was an increase of 14.03% in leaves and 29.3% in fruits, with a consequent reduction in the activity of pectinolytic enzymes, weight loss and fruit rot decay percentage. The fig tree Roxo de Valinhos sprayed with 1.0% $CaCl_2$ presented fruits with larger diameters and greater firmness, as well as higher total phenolic concentrations and greater antioxidant activity for fruit peel and pulp.

$CaCl_2$ pre-harvest sprays are easy management to implement by fig growers at low cost. The main limitations were the number of applications needed to obtain significant changes in the fruit characteristics, and at the same time, without causing symptoms of calcium phytotoxicity on the leaves. Another research plan aims to perform a sensory and volatile compound evaluation of the fruits, in addition to evaluating a smaller number of sprays with $CaCl_2$.

### 5. Conclusions

Pre-harvest sprayings of $CaCl_2$ 1.0% is a promising cultural practice for the production of high-quality fig fruit with greater nutritional value, reducing the activity of pectinolytic enzymes and promoting less weight loss and incidence of rot, i.e., extending shelf-life by approximately 7 to 10 days.

**Author Contributions:** Conceptualisation, S.L. and J.M.A.S.; methodology, S.L., J.M.A.S., M.L. and E.L.G.; validation, J.M.A.S.; formal analysis, J.M.A.S.; investigation, J.M.A.S., E.L.G., L.R.R., R.B.F., R.C.M., M.d.S.S., L.N.H.M. and A.S.D.; resources, S.L. and M.L.; data curation, J.M.A.S.; writing—original draft preparation, J.M.A.S. and S.L.; writing—review and editing, S.L.; supervision, S.L.; project administration, S.L.; funding acquisition, S.L. All authors have read and agreed to the published version of the manuscript.

**Funding:** This work was partially supported by the Brazilian National Council for Scientific and Technological Development (CNPq), grant numbers 302611/2021-5 and 302848/2021-5 and Coordination for the Improvement of Higher Education Personnel (CAPES—PNPD scholarship).

**Institutional Review Board Statement:** Not applicable.

**Informed Consent Statement:** Not applicable.

**Data Availability Statement:** Data are contained within the article.

**Acknowledgments:** We thank Guilherme Augusto Rago Ferraz for providing language assistance.

**Conflicts of Interest:** The authors declare no conflict of interest. The funders had no role in the design of the study; in the collection, analyses, or interpretation of data; in the writing of the manuscript; or in the decision to publish the results.

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
