# Peer review of "Calcium Nutrition in Fig Orchards Enhance Fruit Quality at Harvest and Storage"

_horticulturae, doi:10.3390/horticulturae9010123_

Round 1

Reviewer 1 Report

The research carried out is of great interest to the scientific community.

Methodology

In weight loss, the author does not specify if the same fruits were used for the initial weight and the final weight.

Results

Figure 7 is mentioned in the text, however it is not found in the document.

Author Response

Dear reviewer, we are grateful for your valuable comments. All your comments were considered when adjusting this manuscript. We hope to have answered all of them.

Reviewer Comments/suggestions

Authors corrections/responses

Material and Methods:

In weight loss, the author does not specify if the same fruits were used for the initial weight and the final weight.

The same fruits were used for the initial weight and the final weight. We reframed the sentence in the manuscript.

Results:

Figure 7 is mentioned in the text, however it is not found in the document

Figure 7 was added in the manuscript.

Reviewer 2 Report

The manuscript submitted by Sarita Leonel et al. mainly dealt with calcium nutrition in fig orchards enhancing fruits quality at harvest and storage. The articles was well written and clearly organized. The figures and tables were fitted in the requirements of the journal. However, some problems or errors should be revised according to the following suggestions.

Abstract: please add one sentence at the end of the paragraph to strengthen the significance of this study. 

Introduction

line 67, please add one sentence to intensify the significance of this study.

Materials and methods

line 142 to 174, authors evaluated the TA, reducing,non-reducing and total sugars, GAE, TEAC, why not evaluate the volatile compounds of figs fruits ? And also the sensory evaluation of figs fruits. please make an explanation.

Results

Figure 3, b, there are just 4 points on these lines, please make an explanation.

Figure 4, all curves were depicted from 4 points, were there enough to make a regression ? Please make an explanation.

Figure 5, b, d, e, f , all these curves were just obtained from 4 points, why not 5 points ?

Figure 6, all these curves were just obtained from 4 points, why not 5 points ?

Conclusion

please add the limitations and further research plan of this study.

Author Response

Dear reviewer, we are grateful for your valuable comments. All your comments were considered when adjusting this manuscript. We hope to have answered all of them.

Reviewer Comments/suggestions

Authors corrections/responses

Extensive editing of English language and style required

The authors have requested the service of a specialist to review the English language of the manuscript.  If there is need of any further correction, we kindly ask the reviewer to indicate it to us, and we will do it promptly.

Line 67 – add one sentence to intensify the significance of this study.

The sentence has been added.

Lines 142 to 174 – why not evaluate the volatile compounds of fig fruits? And also the sensory evaluation of fig fruits?

The gas chromatograph for evaluation of volatile organic compounds was broken at the time of fig harvesting.  The protocol for sensory evaluation of the figs was requested from the local ethics committee of the university. However, the response allowing the evaluation took a long time and the fruits could not be evaluated. We should have asked for more notice. This was a learning experience for future research.

Results: Figures 3b, 4 (a and b), 5 (b, d, and f), 6 (a, b, c and d) and 7 (a, b, c and d.  All the curves were just obtained from 4 points, why not 5 points?

In the specified figures referred by the reviewer, in which there was an effect of the interaction between treatments with CaCl2 concentrations and storage times, only 4 storage times were evaluated.  The explanation for this is that although 100 trees were used for the field trial, apart from the 20 threshold trees, there were not enough fruits at the same stage of maturity to evaluate another storage time.

If is necessary we can try to provide more explanations.

Conclusion: please add the limitations and further research plan of this study.

We believe that this is an easy cultural practice to adopt by fig growers and also low cost.  However, our future plans are to decrease the number of CaCl2 applications during fruit development, which were necessary to achieve the results obtained in the first years, and also to perform sensory evaluation in addition to volatile compounds.  A sentence has been added at the end of the discussion.

Reviewer 3 Report

The authors study pre-harvest applications of calcium chloride in fig orchards at different CaCl2 concentrations and fruits were stored from 0 to 21 days. They analyzed the content of calcium in leaves and fruits, weight loss, fruit decay, pectinolytic enzyme activity, physical and chemical attributes of the fruits. The manuscript is well written; however, some revisions should be done to improve the quality of it.

Points that should be reviewed by the authors:

·        Abstract: Line 18. In what stage of maturation calcium was applied?

Line 19. At ambient temperature?

Line 21. It increases, what percentage?

Line 24-25. You should include a real conclusion obtained with the results of your works. It has not been indicated how many days the shelf-life of the product in storage is extended, only that it improves certain parameters.

In keywords you mention phenols, but in abstract you do not include any relevant data only Line23-24.

·        Introduction: The introduction part is well structured.

Line 46-50. The reference 8 and 9 are old from 2011 and 2014. There are more current references. Expand this paragraph in depth.

I have carried out a bibliographical search and there are much more recent works in the line of work that they have carried out. For instance, you should take into account the following references. You can perfectly justify the effect of precalcium treatment in phenol and AA in olives trees and even in the table olives elaborated. You can even discuss your results with these references.

-Lodolini, E. M., Fernández, A., Morales-Sillero, A., Mendiano, A., & Martín-Vertedor, D. (2023). Influence of pre-harvest calcium applications on table olive characteristics during Spanish-style elaboration process. Scientia Horticulturae, 308, 111577.

-Morales‐Sillero, A., Lodolini, E. M., Suárez, M. P., Navarrete, V., Jiménez, M. R., Casanova, L., ... & Martín‐Vertedor, D. (2021). Calcium applications throughout fruit development enhance olive quality, oil yield, and antioxidant compounds' content. Journal of the Science of Food and Agriculture, 101(5), 1944-1952.

Line 61-62. In Californian-style black and green olives, the effect of calcium strategies by immersion the fruit is well-known. You can include these references. You can also include in this paragraph something about that the immersion of the fruit in calcium can provide some bitterness negative attributes provided by this mineral.

Martín-Vertedor, D., Fernández, A., Mesías, M., Martínez, M., Díaz, M., & Martín-Tornero, E. (2020). Industrial strategies to reduce acrylamide formation in Californian-style green ripe olives. Foods, 9(9), 1202.

Martín-Vertedor, D., Fernández, A., Mesías, M., Martínez, M., & Martín-Tornero, E. (2021). Identification of mitigation strategies to reduce acrylamide levels during the production of black olives. Journal of Food Composition and Analysis, 102, 104009.

Line 63-64. I completely agree with you. You may extent this part.

·        Materials and Methods:

Line 85. Why do you use this adhesive?

Line 86. 10 applications of calcium. It seems excessive calcium application. ESTADO DE Line 92. You do not disinfect previously the figs?

Lines 176-180. Guard trees were used to preserve some treatments from others

The fruits in what state of maturation they were collected

Why was a sensory analysis of the fruits not carried out to verify if high concentrations of calcium applied and with so many applications carried out could affect the bitterness of the treated fruit.

Do you think that this assay affects the sensory quality of the final product?

Some type of caking agent was used to ensure that the calcium was adhered to the fruit and leaves.

·        Results: It is well-written and exposed. Results are detailed and well discussed.

Line 201. Write well CaCl2.

Line 207-208. I think that the concentration of calcium is not excessively high but you have applied a lot of treatments during the vegetative cycle.

Have you checked if there is any correlation between agronomic and chemical variables carried out in the different experimental treatments studied?

·        Discussion: The discussion part it seems like an introduction part. You should explain part of the results compared with the references. I would start the discussion part with general information and after that I will compare your results with other previously done.

As I mentioned before, you should reference the results found in your work with others in the bibliography. Both works carried out with olives and pre-calcium treatments can help you to discuss this part.

Why do you think it increases antioxidant activity and phenol content?

and the weight of the fruit?

It would be necessary to make a small “conclusion” or summary of the results found discussed in this article.

·        Conclusions: It could be interesting if you can include a general conclusion relate to the practical aspect of this study at industrial level and the relevance of the study.

A general conclusion would be missing on the best recommendation to make to preserve the fruit with the highest possible quality.

What is the best concentration of calcium that makes the fruits last longer with the highest possible quality?

·        References: Some bibliographical citations are too old. Try to put more current cites related to the theme of the work.

Author Response

Dear reviewer, we are grateful for your valuable comments. All your comments were considered when adjusting this manuscript. We hope to have answered all of them.

Reviewer Comments/suggestions

Authors corrections/responses

Abstract:

Line 18 – In what stage of maturation calcium was applied?

An additional sentence has been added in the manuscript. Calcium was applied when fruits emission began and finished in a ripening stage less unripe, with firmer fruits.

Line 19 – At ambiente temperature?

Yes. The sentence has been added in the manuscript.

Line 21. It increases, what percentage?

CaCl2 sprays enhanced fruit Ca2+ concentration in leaves (14.03%) and fruits (29.3%). The sentence has been added in the mansucript.

Line 24-25. You should include a real conclusion obtained with the results for your works. It has not been indicated how many days the shelf life of the product in storage is extended, only that it improves certain parameters.

The pre-harvest application at 1% CaCl2 showed to be a promising technique in producing high quality fruits and extended storage in approximately 7 days. The sentence has been added in the mansucript.

Keywords:

You mention phenols, but in abstract you do not include any relevant data only line 23-24.  

We removed the keyword phenolic compounds.

Introduction:

Line 46-50. The reference 8 and 9 are old from 2022 and 2014. “...You can even discuss your results with these references”.

The references 8 and 9 were reframed according with the references send by the reviewer.

Line 63-64. You may extent this part.

This part has been extended.

Material and Methods:

Line 85. Why do you use this adhesive?

We used Assist adhesive spreader to improve the absorption with the calcium treatments. We obtained positive results with the use of this application system in guava.  Ribeiro, L.R.; Leonel, S.; Souza, J.M.A.; Garcia, E.L.; Leonel, M.; Monteiro, L.N.H.; Silva, M.S. Improving the nutritional value and extending shelf life or red guava by adding calcium chloride. LWT-Food Science and Technology. 2020. 130, 109655.

Line 86. 10 applications of calcium. It seems excessive calcium application.

Line 92. You do not disinfect previously the figs?

Lines 176-180. Guard trees were used to preserve some treatments from others?

The fruits in what state of maturation where collected?

Why a sensory analysis of the fruits not carried out to verify it high concentrations of calcium applied and with so many applications carried out could affect the bitterness of the treated fruit?

Do you think that this assay affects the sensory quality of the final product?

We have plans to try to decrease the number of CaCl2 applications in future research. However, it is very important to point out that there is a lack of information about the source, concentration, quantity and period of calcium application in fig trees to obtain an effective response of fruit quality.

The figs were previously disinfected.

2.3 …After harvesting, they were cleaned in running water and placed on benches for natural drying.

Yes. Twenty guard trees were used to preserve some treatments from others. One sentence has been added in the manuscript.

Calcium chloride was applied when fruits emission began and finished in a ripening stage less unripe, with firmer fruits [19].

The protocol for sensory evaluation of the figs was requested from the local ethics committee of the São Paulo State University. However, the response allowing the evaluation took a long time and the fruits could not be evaluated. We should have asked for more notice. This was a learning experience for future research.

We do not think that this assay affected the sensory quality of the final product in terms of bitterness, even though it was not evaluated. On the contrary, treatment with 1.0% CaCl2 effectively improved fig fruit quality.

Results:

Line 201. Write well CaCl2

OK.

Line 207-208. I think that the concentration of calcium is not excessively high but you have applied a lot of treatments during the vegetative cycle.

Calcium is moved in the plant metabolism by the transpiration flow and part of the calcium reallocated in the fruits ends up being translocated to meristematic regions where there are higher transpiration rates, such as growing leaves, this fact highlights the need for a constant supply of this nutrient during fruit growth.

We have already tried to clarify this issue in the manuscript.

Have you checked if there is any correlation between agronomic and chemical variables carried out in the different experimental treatments studied?

We did not assess agronomic variables in this study.

Discussion:

Why do you think it increases antioxidant activity and phenol content?

And the weight of the fruit?

Calcium promotes the activation of the enzyme NADPH-oxidase, which increases the production of reactive oxygen species (ROS) (O2 and H2O2). Due to the increased level of ROS, the activity of the enzyme phenylammonium lyase (PAL) is increased, thus promoting the synthesis of phenolic compounds.

Calcium plays an important role in changes in cell turgor pressure and cell elongation in fruits, in addition to being involved in auxin signaling and acidification of the cell wall, allowing its expansion, which can lead to increased fruit weight. However, the reduction in averages during storage, whose was observed in all treatments, is related to the increase in weight loss in this period.

These possible explanations were presented in the manuscript.

Conclusions:

It would be interesting if you can include a general conclusion to the practical aspect of this study at industrial level and the relevance of the study.

The conclusion was reframed considering the reviewer suggestions.

References:

Some bibliographical citations are too old. Try to put more current citations related to the theme of the work.

The new citations were added to the work according with the reviewer’s suggestion.

Reviewer 4 Report

Dear Authors, 

Please, refer to the comments/suggestions given in the pdf manuscript.

Thank you

Author Response

Dear reviewer, we are grateful for your valuable comments. All your comments were considered when adjusting this manuscript. We hope to have answered all of them.
The corrections presented in the yellow balloons were carried out in the manuscript.

Reviewer 5 Report

The manuscript proposed by Jackson Mirellys Azevêdo Souza and his colleagues reports the pre-harvest CaCl2 sprays as a technology to extend the storage and calcium content of figs.

 This manuscript matched the scientific scope of the Horticulturae  journal and can be accepted after major revisions. The suggestions are listed below:

-        Subscript CaCl2 at paragraph 201, 218, 238, 256 and 273

-        Replace Figure 3A with Figure 3a

-        Replace Figure 3B with Figure 3b

-        Subscript at the legend of Figure 4, Figure 5 and Figure 6

-        Figures 7a-7d are missing from the manuscript.

Author Response

Dear reviewer, we are grateful for your valuable comments. All your comments were considered when adjusting this manuscript. We hope to have answered all of them.

Reviewer Comments/suggestions

Authors corrections/responses

Subscript CaCl2 at paragraph 201, 218, 238, 256 and 273

 - Replace Figure 3A with Figure 3a

- Replace Figure 3B with Figure 3b – Subscript CaCl2 in figures.

- Figures 7a-7d are missing from the manuscript.

The corrections has been made.

Round 2

Reviewer 2 Report

Accept

Reviewer 3 Report

The authors have made a number of changes to the text that significantly improve the quality of the publication. The language, references and technical aspects used in the article has been improved. Thoughts and facts were exposed clearly. The well-written manuscript about has a significant relevance to food science research field. Therefore, I believe that all necessary corrections have been made by the authors and the article should be considered for publication in the Journal. I have no comments on the current content of the article. My sincere congratulations to the authors for their efforts and commitment with the scientific community.

Reviewer 5 Report

Agree with the revision manuscript.